# Test-retest reliability of the six-minute walking distance measurements using FeetMe insoles by completely unassisted healthy adults in their homes

**Andrey Mostovov**[1]*, **Damien Jacobs**[1], **Leila Farid**[1], **Paul Dhellin**[1], **Guillaume Baille**[2]

**1** FeetMe SAS, 157 bd. MacDonald, Paris, France, **2** Neurology department, Delafontaine Hospital Center, Saint-Denis, France

* science@feetme.fr

**Data Availability Statement:** The data that support the findings of this study are publicly available from

## Abstract

Wearable technology provides an opportunity for new ways of monitoring patient gait remotely, through at-home self-administered six-minute walk tests (6MWTs). The purpose of this study was to evaluate the test-retest reliability of FeetMe insoles, a wearable gait assessment device, for measuring the six-minute walking distance (6MWD) during tests conducted with a one-week interval by completely unassisted healthy adults in their homes. Participants (n = 21) performed two 6MWTs at home while wearing the FeetMe insoles, and two 6MWTs at hospital while wearing FeetMe insoles and being assessed by a rater. All assessments were performed with a one-week interval between tests, no assistance was provided to the participants at home. The agreement between the 6MWD measurements made at baseline and at Week 1 was good for all test configurations and was highest for the at-home FeetMe measurements, with an intraclass correlation coefficient (ICC) of 0.95, standard error of the measurement (SEM) of 15.02 m and coefficient of variation (CV) of 3.33%, compared to ICCs of 0.79 and 0.78, SEMs of 25.65 and 26.65 and CVs of 6.24% and 6.10% for the rater and FeetMe measurements at hospital, respectively. Our work demonstrates that the FeetMe system could provide a reliable solution allowing individuals to self-administer 6MWTs independently at home.

## Author summary

At-home patients monitoring using wearable tools presents numerous advantages for regular care and clinical studies. Patients benefit from the convenience of not having to travel to a clinic for assessments, which is particularly helpful for those in remote areas or with mobility issues. Besides, at-home monitoring allows for more frequent assessments, leading to more accurate clinical decision-making and timely intervention, which ultimately results in enhanced patient care. For the same reasons, it can improve patients recruitment and retention in clinical studies. The six-minute walk test (6MWT) is a commonly used standardized assessment of functional capacity in patients with various diseases. We

Zenodo public repository with the identifier https://doi.org/10.5281/zenodo.8252988.

**Funding:** The authors received no specific funding for this work.

**Competing interests:** I have read the journal's policy and the authors of this manuscript have the following competing interests: A.M. is employee of the FeetMe® company, D.J. and L.F. were employees of the company at the time of the study and P.D. was the company's intern at the time of the study. G.B. has no conflicts of interest to disclose and has no link of interest to the FeetMe® company.

evaluated the test-retest reliability of FeetMe insoles, a wearable gait assessment device, for measuring the six-minute walking distance (6MWD) at home. Our analysis of the data from 21 healthy volunteers showed that the FeetMe insoles were as reliable at home as they were in the standard clinical setting and as a conventional way of assessing the 6MWD. In addition, the insoles provided extensive gait analysis, which may allow for more precise conclusions regarding the patient's state and its evolution than the 6MWD alone. We conclude that the FeetMe device is an excellent candidate tool for at-home patients monitoring.

## Introduction

Gait is considered as a reliable indicator of overall health status. A range of conditions, such as neurological diseases, can lead to gait impairment including a slow gait speed, gait asymmetry, and an unbalanced center of gravity [1,2]. Several tests have been developed to evaluate these parameters, such as the two-minute walking test (2MWT) [3], the six-minute walking test (6MWT) [4], and the twelve-minute walking test (12MWT) [5]. Among these tests, the 6MWT is the most widely used assessment and has emerged as the "go-to" gait evaluation test in clinical practice. It is easy to administer and well tolerated, and has been found to provide a better reflection of a patient's capacity for daily physical activity than other tests [4]. The test involves measuring the distance walked by a patient during a 6-minute time frame (i.e., the six-minute walking distance; 6MWD). It has been used conclusively in many clinical investigations. The test-retest reliability of the 6MWT has been found to be excellent, with reported intraclass correlation coefficients (ICCs) of 0.91–0.98 and inter- and intra-rater reliability ICCs of 0.86–0.96 [6–8].

Despite its many advantages, the 6MWT does have some limitations that need to be addressed. First, the test is typically carried out and monitored manually by a rater, which may lead to inconsistences in the assessment. Most notably, words of encouragement [9] or variations in the instructions provided by the rater [10] could impact the patient's performance. Second, the test only evaluates a single gait parameter (the average gait speed over 6 minutes); whereas other gait parameters, such as stride length or stance time, have been shown to be useful for evaluating patient health status. Indeed, such gait parameters have been shown to be valuable indicators of fatigue in patients with multiple sclerosis [11].

Given the widespread use of the 6MWT in clinical practice for evaluating the functional capacity of patients with a range of diseases, there is great interest in developing technical solutions that could help to streamline the evaluation process, allowing tests to be conducted more frequently and provide more detailed follow-up data and improved care. This demand aligns with the advancements of wearable technology, which present new opportunities for improving patient care and monitoring through innovative means. In particular, exploring whether wearable devices could be used to allow gait assessments to be conducted at home could be highly beneficial for patients. Self-administering gait tests at home could eliminate the patient burden associated with commuting to clinical facilities. At-home monitoring would also allow for repeated tests to be conducted over an extended period of time, allowing physicians to gather more longitudinal data and improve patient follow-up. Several research groups have proposed ways to assess gait parameters in the comfort of the patient's own home and have evaluated the interest of such solutions. In one of the early studies, Alison et al. highlighted the interest of performing the 6MWT at home in survivors of critical illness [12]. However, in this study the 6MWTs were still administered in the conventional manner, with the rater being

dispatched to the patient's home to administer the test, something that is seldomly possible outside of a clinical study. Studies testing the reliability of an accelerometer-based quantification program [13] and a wearable guided 6MWT device [14] to measure gait speed in patients with cardiovascular conditions have obtained promising results, with the at-home distances measured with these devices being consistent with those obtained manually with clinical guidance. Such tools designed for at-home use are therefore likely to play an increasingly important role in the follow-up of patients with gait disorders. Still, only a few of these studies have involved devices that could provide a complete solution, allowing patients to conduct 6MWTs independently, without supervision by a healthcare professional, and provide estimates of the 6MWD without any assumptions or a priori information.

FeetMe insoles are among the recently developed wearable devices that have the capability to allow gait assessments to be self-administered remotely in the patient's own home. These insoles were designed to assess many gait parameters, including stride length, velocity, stance, swing, step, single and double support durations, and cadence and their reliability have already been proven by independent studies. Test-retest reliability of gait parameters was confirmed in older adults [15] and the insoles were found to be as reliable as the GAITRite clinical walkway system for walking tests conducted in clinical settings in patients with multiple sclerosis [16] and Parkinson's disease [17]. Other studies in patients with post-stroke [18] and in healthy volunteers [19] came to similar conclusion. The FeetMe insoles have also already been found to be a valid and accurate solution for measuring the 6MWD in hospital settings when compared to the ground truth measured by a surveyor's wheel and estimates made by a rater [20]. The aim of the current study was to probe the test-retest reliability of FeetMe insoles in estimating the 6MWDs in a home environment. This was accomplished by assessing the repeatability of the 6MWTs completed by unassisted healthy volunteers in their homes with a one-week interval between the tests.

## Materials and methods

### Study design

This single-center, prospective study was conducted between October 2021 and August 2022 by investigators working at the Delafontaine Hospital Center (Saint-Denis, France). The study was approved by a French ethics committee, CPP EST I, and complied with the Declaration of Helsinki and all subsequent amendments (registration number, ID-RCB: 2021-A00037-34).

The primary aim of the study was to assess the reliability of 6MWD measured using connected insoles in a remote home setting along a 10-meter track. Additionally, the study aimed to compare the reliability of this remote measurement to the reliability of the conventional assessment conducted in a hospital setting, where both the insoles and a rater measured the 6MWD simultaneously also along a 10-meter track.

### Inclusion

All healthy volunteers aged between 18 and 80 years old, who were able to walk 100 m unaided, had no gait disorders, and who were accustomed to using a smartphone, were eligible to participate in the study. Volunteers who had undergone surgery that could potentially impact gait in the previous 3 months (e.g., orthopedic surgery, an intervention for trauma of the lower limbs or spine, gynecological or urological surgery, or brain or spinal cord surgery) and those with a chronic disease affecting walking (e.g., rheumatological, orthopedic, pain, or neurological disorders) were excluded. The participants were recruited using study posters displayed in the relevant areas (e.g. universities for the younger age groups, typical associations for older age groups). The volunteers were provided with information about the study by phone or e-

mail prior to the study start, and were given the opportunity to ask any the questions. All volunteers provided signed consent prior to the study start.

## Instrumentation

The study used size 35 to 46 FeetMe insoles (FeetMe SAS, Paris, France), a Class Im CE(93/42/EC) and Class I FDA 510(k) exempt medical device (Fig 1). The technical characteristics of the insoles have been described previously [19]. The FeetMe insoles were used together with the FeetMe Evaluation smartphone application (Class Im CE(93/42/EC) and Class I FDA 510(k) exempt medical device) to administer the 6MWT. Data collected by the FeetMe insoles were transferred to the smartphone application via a Bluetooth Low Energy (BLE) emitter in the insoles, allowing information on plantar pressure, gait parameters and walking distance to be received in real time. Users selected and launched the 6MWT through the smartphone application. Once the 6MWT had been launched, the application collected and recorded the user's gait parameters for each of their steps over the entire duration of the test, and then automatically stopped recording after 6 minutes and informed the user that the test had been completed. The data then were automatically sent to a medical service-certified secure server and test results could be displayed in the application or on the associated web platform, the FeetMe Mobility Dashboard. The access to the data and to the test results is fully restricted to authorized persons only. The data was handled in accordance with the French data protection authority's (CNIL) reference methodology MR-003 as well as in accordance with the requirements of the European Union General Data Protection Regulation (GDPR).

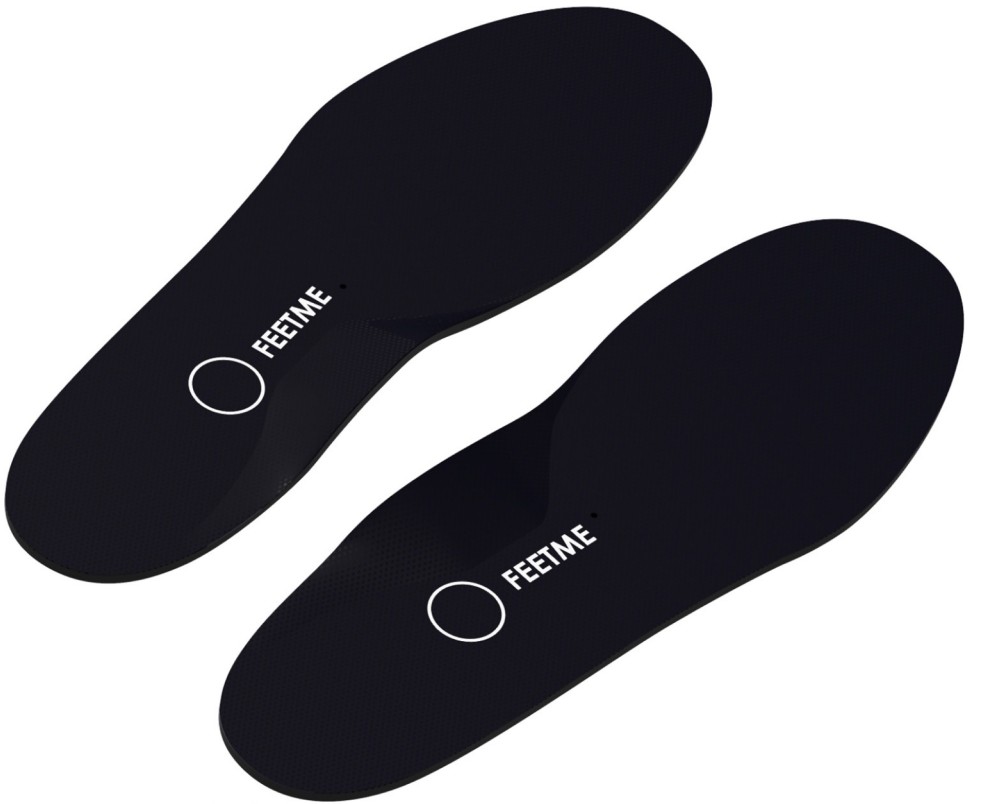

**Fig 1. A pair of FeetMe Insoles.**

For the 6MWTs performed at home, participants were provided with simple equipment—two hoops (diameter: 0.5 m) attached together with a 10-m string—to allow them to define a 10-m track in their home surroundings.

## Intervention

Prior to the intervention, all volunteers were given a 5 to 10 minutes training on the use of the FeetMe insoles and FeetMe Evaluation smartphone application so that they could use the system independently, without any support from a nurse or other healthcare professional. The investigator also helped selecting the insoles size and insured that they fit well participants' shoes. The healthy volunteers then wore the insoles while carrying out 6MWTs at two hospital visits (baseline: day 0, and Week 1: day 8) and on two occasions at home (baseline: day 1, and Week 1: day 7). In hospital, the participants were evaluated by a rater (a medical doctor or an engineer) trained to administer the 6MWT according to the study protocol.

For the two tests conducted at the hospital, data were analyzed for each participant as they performed the 6MWTs walking at a comfortable speed on a 10-m track, while wearing the FeetMe insoles and being simultaneously assessed by a rater. Contrary to the official test guidelines [21], no signs of encouragement were provided by the rater during the test. The rater only informed the participant of the time remaining every minute, then 30 seconds and 10 seconds before the end of the test.

For the two tests conducted at home, volunteers were provided with the insoles, a smartphone with the FeetMe Evaluation application, and the track equipment. They were asked to perform the 6MWTs while wearing the insoles on a 10-m track made using the equipment provided. The test could be performed in a quiet place either indoors or outdoors (undercover if required by weather conditions), but required a flat and hard surface, with few or no passages and, ideally, no obstacles.

## Outcomes

The main outcome was the test-retest reliability of the FeetMe 6MWD measurements from tests performed by the participants at home without any assistance. The test-retest reliability of the 6MWD measurements made by the rater in hospital and by FeetMe in hospital in the same population was used as a reference level of repeatability.

## Statistical analysis

The normality of the 6MWD data was assessed using Q–Q plots and Shapiro-Wilk normality tests. The mean and standard deviation (SD) of the recorded 6MWDs were calculated for the rater at hospital, FeetMe at hospital and FeetMe at home at baseline and Week 1. The bias (i.e., systematic error), the 95% confidence interval (CI) of differences (i.e., limits of agreement), Pearson correlation coefficient, ICC (2,1), coefficient of determination, standard error of the measurement (SEM) and coefficient of variation (CV) were calculated to compare baseline test results with test results obtained at Week 1 for each of the three test configurations. A Levene test was used to assess significant differences between the SDs of the test results at baseline and at Week 1.

The repeatability of the test results obtained at the two timepoints was analyzed using Bland-Altman and linear regression plots for all three configurations: rater at hospital, FeetMe at hospital and FeetMe at home.

The following criteria were used to assess the degree of correlation [22]: <0.30 negligible, 0.30–0.50 low, 0.50–0.70 moderate, 0.70–0.90 good, and 0.90–1.00 excellent. The same criteria were used for the coefficients of determination. For the ICCs, values below 0.50 were deemed

to indicate poor validity, values between 0.50 and 0.75 to indicate moderate validity, values between 0.75 and 0.90 to indicate good validity and values greater than 0.90 to indicate excellent validity (as described previously [23]). A priori significance levels ($\alpha$) were set at 0.05 for all analyses. All data and statistical analyses were performed using Python software (version 3.8).

## Results

### Demographics and population distribution

A total of 33 healthy volunteers, 15 females and 18 males, were included in the study. Participants ranged in age from 23 to 73 years, with a mean of 42 years. The average height and weight of the population were 173.9 ± 9.3 cm and 70.9 ± 10.9 kg, respectively.

### Results of the tests

Overall, 30 out of the 33 participants completed all the tests in the hospital setting (Fig 2). Among these 30 participants, one participant performed no tests at home and four performed the test at home on only one out of the two days required or performed an incorrect type of test on one of the days. In addition, three of the participants carried out tests that were shorter than 6 minutes, and a technical issue (application crash) prevented data from being recorded in one case. Data from all of these participants were excluded from the analysis and therefore the final analysis population consisted of 21 healthy volunteers.

Q-Q plots evaluating the normality of the data for each test configuration (rater at hospital, FeetMe at hospital and FeetMe at home) both at baseline and at Week 1, indicated that the distribution of the data was close to normal in all cases (Fig 3). The Shapiro-Wilk tests yielded p-values greater than 0.05 for the data collected at baseline for all three configurations. However, Shapiro-Wilk p-values for the data collected at Week 1 by the rater and by FeetMe at home were equal to or below 0.05 (Table 1).

### Repeatability assessments

The mean and SD of the 6MWDs measured at baseline were very similar to those estimated at Week 1 for the FeetMe evaluations conducted at home. On the contrary, when comparing the measurements taken during the initial hospital visit with those of the second visit, we noted a notable disparity in both the mean and SD estimators (Table 1). This observation held true for both the rater assessments and the FeetMe assessments. The results of the Levene test showed that there were no significant differences in the SDs of the distance estimates made at baseline and Week 1 for all three test configurations (Table 1).

The FeetMe at home measurements showed a very low test-retest bias of 2.15 m, which was less than 0.5% of the total distance measured at both time points (Table 2). In comparison, the equivalent bias for the rater assessment was -14.57 m. The results of the linear regression analysis between the two visits for each test configuration (rater at hospital, FeetMe at hospital and FeetMe at home) are shown in Fig 4. It is noticeable that for FeetMe at home estimates, the regression line was very close to the ideal reference. This observation was further confirmed by the coefficient of determination value of 0.90 and the Pearson correlation coefficient value of 0.95 indicating an excellent level of correlation between the 6MWDs measured at the two timepoints (Table 2). In addition, the 95% CIs of the ICC (0.88–0.98) for the at-home FeetMe measurements indicated a very good to excellent intraclass correlation between the distances measured by the device at the two timepoints. By contrast, the coefficient of determination value of 0.66 and the Pearson correlation coefficient value of 0.81 indicated that the

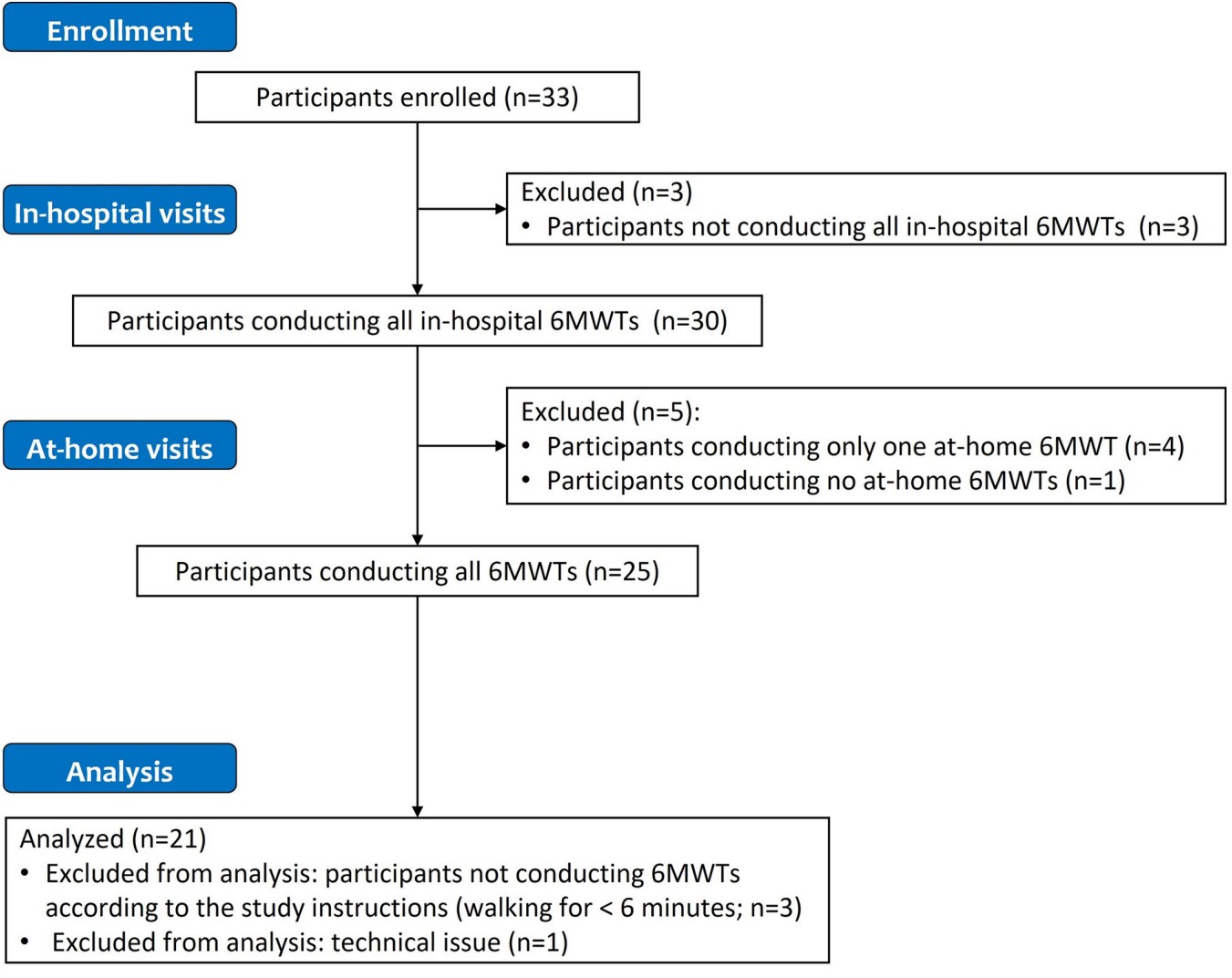

**Fig 2. Flowchart of participant enrolment and data exclusion.** Abbreviation: 6MWT, six-minute walking test.

measurements made by the rater showed a moderate to good correlation between the two timepoints, and the 95% CIs of the ICC (0.54–0.91) indicated a moderate to excellent intraclass correlation for the rater measurements. The SEM and the CV values were also lower for the measurements made during the FeetMe at-home assessments than for those made during the rater assessments at hospital: 15.02 m and 3.33% for FeetMe at home versus 25.65 m and 6.24% for the rater, respectively. Analysis of the Bland-Altman plots (Fig 5) confirmed these conclusions: in addition to showing much lower bias, the FeetMe at home estimates showed substantially narrower limits of agreement compared to those for the two in-hospital test configurations.

## Discussion

This study evaluated the potential of FeetMe insoles, a connected wearable gait assessment device, to provide a solution allowing 6MWTs to be self-administered, independently and accurately, by individuals in their own home. The test-retest reliability of the 6MWDs

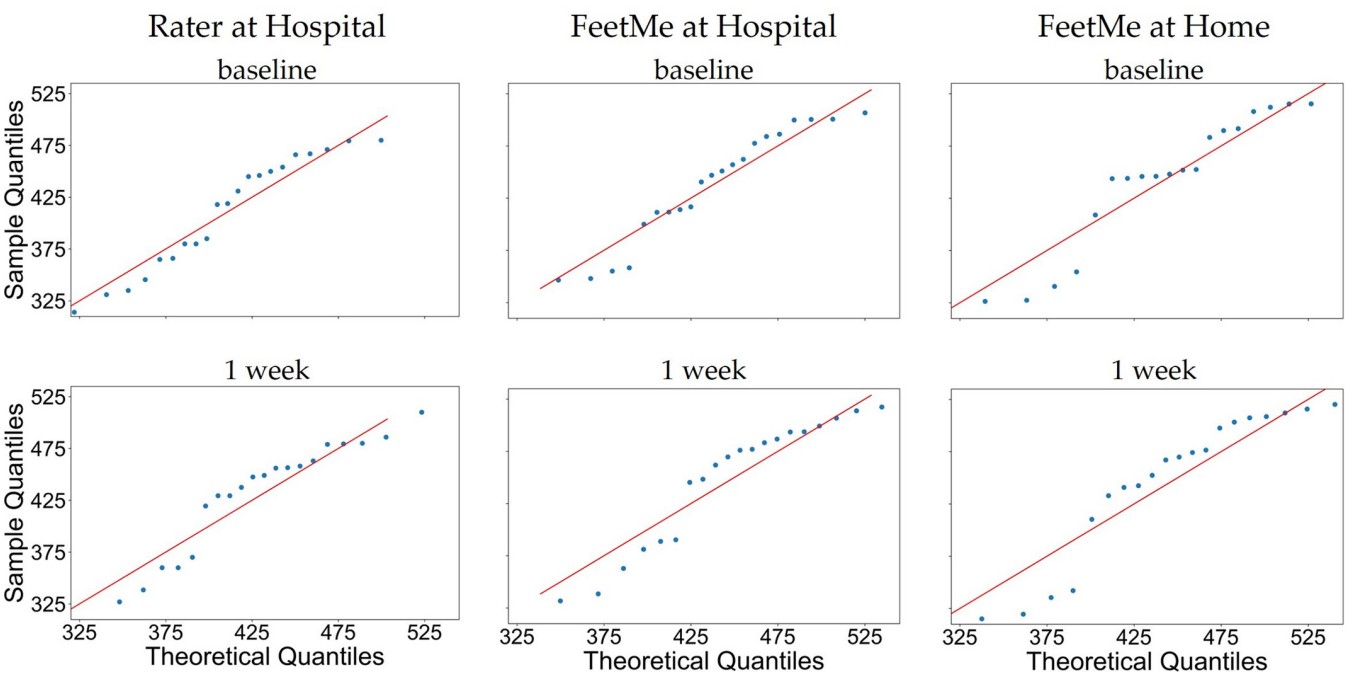

**Fig 3. Q-Q plots for the 6MWDs evaluated at baseline and Week 1 by a rater at hospital, and by FeetMe at hospital and at home.**

measured by the FeetMe insoles during two at-home 6MWTs performed within a one-week of interval by unsassisted healthy participants was compared with that for measurements obtained in hospital by a rater and using the FeetMe insoles. Our study demonstrated that, while there was good agreement between the test-retest measurements for all three test configurations, the 6MWD measurements made by the FeetMe insoles at home had higher ICC and coefficient of determination values, and lower bias, SEM and CV% values than those obtained for the in-hospital FeetMe and rater measurements. Furthermore, the learning effect [24], which was evident in the in-hospital measurements, was not observed in the measurements conducted at home. This difference suggests that the FeetMe insoles facilitate consistent and unbiased assessments during at-home testing, unaffected by the learning curve observed in the hospital setting. Thus, at-home self-administered 6MWTs using the FeetMe technology were found to be at least, if not more, reliable than rater assessments conducted in a hospital setting, providing evidence that the FeetMe insoles could deliver an easy-to-use, reliable, and accurate solution allowing patients to conduct 6MWTs at home.

**Table 1. Mean and standard deviation of the 6MWD measurements obtained at baseline and at Week 1 by the rater and FeetMe at hospital and by FeetMe at home.**

| Test configuration | n | 6MWD at baseline | | 6MWD at Week 1 | | Shapiro-Wilk p-values* | | Levene test p-values** |
|---|---|---|---|---|---|---|---|---|
| | | Mean [m] | SD [m] | Mean [m] | SD [m] | Baseline | Week 1 | |
| Rater at hospital | 21 | 410.86 | 52.60 | 425.43 | 57.87 | 0.09 | 0.05 | 0.82 |
| FeetMe at hospital | 21 | 436.93 | 52.14 | 453.20 | 61.09 | 0.07 | 0.06 | 0.77 |
| FeetMe at home | 21 | 453.02 | 67.18 | 450.87 | 67.02 | 0.09 | 0.004 | 0.98 |

*The Shapiro-Wilk test was used to assess the normality of the data obtained from the three configurations at baseline and at Week 1.

**The Levene test p-values were obtained from comparisons between the SDs of the test results at baseline and Week 1. Abbreviations: 6MWD, six-minute walking distance; n, number of participants; SD, standard deviation.

**Table 2. Analysis of the test–retest reliability of the 6MWD measurements at baseline and Week 1 by the rater, and by FeetMe at hospital and FeetMe at home.**

| Test configuration | n | Bias [m] | Limits of agreement [m] | Coefficient of determination | Pearson Correlation | ICC [lower–upper 95% CI] | SEM [m] | CV [%] |
|---|---|---|---|---|---|---|---|---|
| Rater at hospital | 21 | -14.57 | [-81.83–52.69] | 0.66 | 0.81 | 0.79 [0.54–0.91] | 25.65 | 6.24 |
| FeetMe at hospital | 21 | -16.27 | [-85.29–52.74] | 0.67 | 0.82 | 0.78 [0.53–0.91] | 26.65 | 6.10 |
| FeetMe at home | 21 | 2.15 | [-40.24–44.54] | 0.90 | 0.95 | 0.95 [0.88–0.98] | 15.02 | 3.33 |

Abbreviations: CI, confidence interval; CV, coefficient of variation; ICC, intraclass correlation coefficient; SEM, standard error of the measurement.

The performance of FeetMe insoles in this study was assessed in a population of 21 healthy volunteers, aged between 23 and 73 years old. The age range of the study population was therefore wide enough to cover various levels of physical performance. The excellent test-retest ICC value obtained in our study for the at-home FeetMe measurements (0.95, 95% CI: [0.88–0.98]) was similar to the ICC values reported previously in the literature for repeated 6MWTs conducted in controlled settings with conventional assessment of the 6MWD by a rater (e.g. 0.98, 95% CI: [0.97–0.99] [6] and 0.93 [25]). In contrast, the ICC values obtained for the measurements made by the rater (0.79, 95% CI: [0.54–0.91]) and by the FeetMe device in hospital (0.78, 95% CI: [0.53–0.91]) were lower than those reported previously. The CV% values obtained in our study were lower than those reported previously (e.g., 8% [26]) for all three test configurations studied. This difference might be explained by the fact that previous studies involved different populations (frail older adults with dementia [6], patients with chronic obstructive pulmonary disease, COPD [7] or patients with osteoarthritis [26]) with different age ranges from those used in our study, or conducted the assessments on tracks with different lengths, following the official test guidelines. Indeed, the track length of 10 m used in our study was optimized to allow for the test to be conducted in all home environments with minimal equipment. However, the same space constraints may also occur in hospital settings where use of the 30-m track recommended in the official test guidelines is not always feasible. Our study therefore also provides strong evidence validating the use of shorter track lengths in all settings, which should be considered in any future revisions of 6MWT guidelines.

Besides, it is of interest to compare our findings with the results reported in previously published research. A few of the solutions assessed in previous studies used accelerometer signals

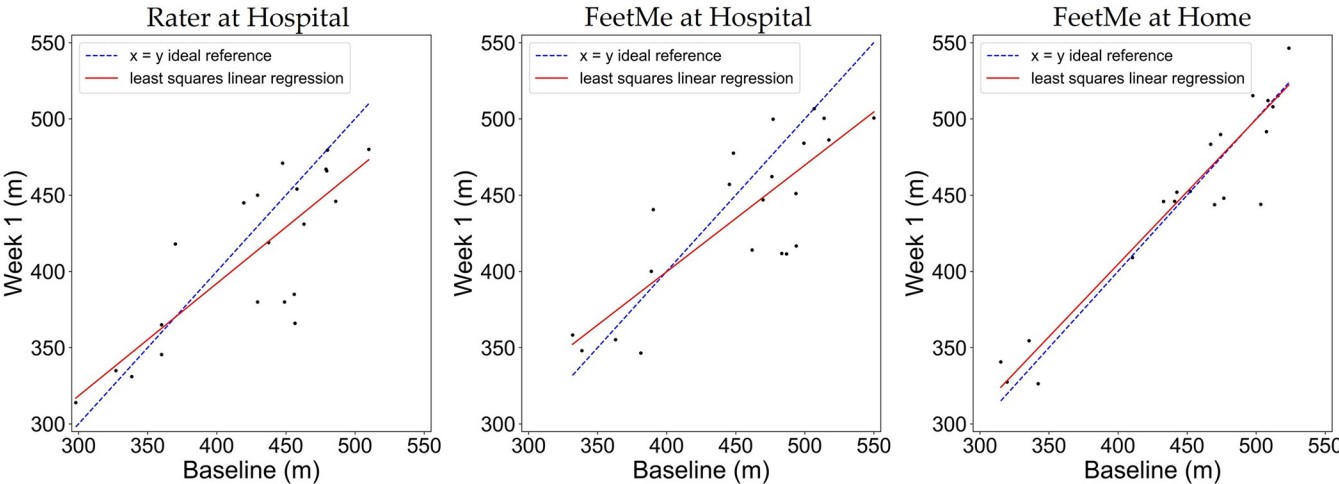

**Fig 4. Linear regression plots between the 6MWD evaluated at baseline and Week 1 by the rater at hospital, and by FeetMe at hospital and at home.** Red lines denote the linear regression, and dashed blue lines indicate the line of ideal match between the Baseline and Week 1 measurements.

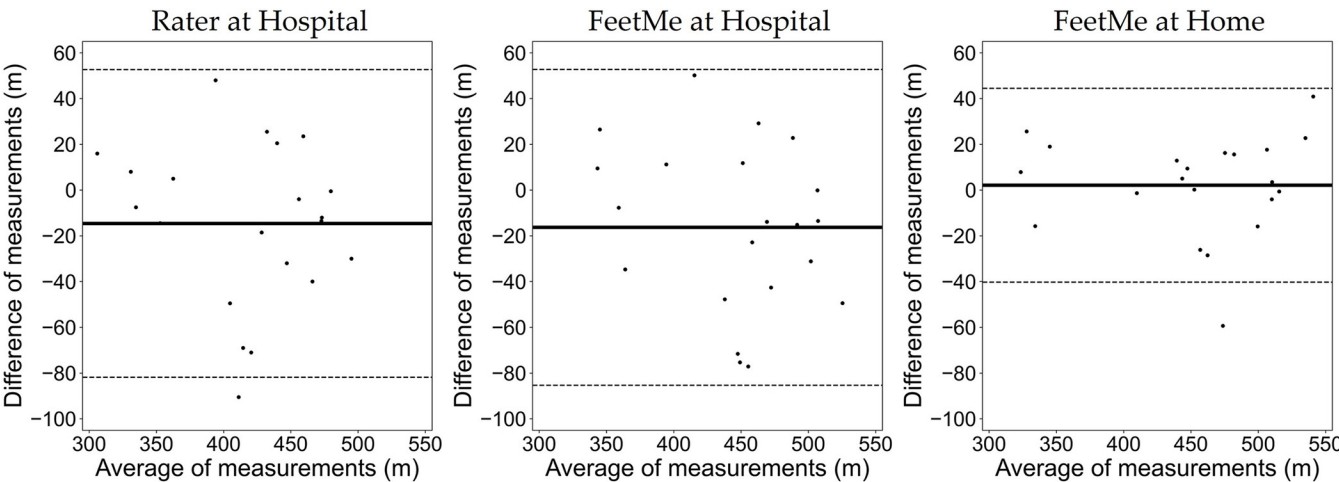

**Fig 5. Bland-Altman plots between the 6MWD estimated at baseline and at Week 1 by the rater at hospital, and by FeetMe at hospital and at home.** The solid lines indicate the bias (mean difference) values, and the dashed lines indicate the upper and lower limits of agreement (95% confidence intervals).

to measure gait and focused only on the number of steps taken [13,14,27,28]. These studies either did not include any assessment of the 6MWD, or tried to derive the distance walked based on a priori information such as patient height or average stride length at baseline. However, these derived estimates are prone to error as stride length has been shown to vary over time in patients with pathologies such as stroke, cerebral ataxia or Parkinson's disease, especially after the patient has received physical therapy [29–31].

In a preliminary study conducted in a laboratory setting, Smith-Turchyn et al. evaluated the potential of the EasyMeasure application as an aid for self-administering 6MWTs at home [32]. As part of the experimental design, the participants were responsible for timing the test and had to manually count the laps walked. Although this study was carried out using a healthy population of 20 young university student volunteers, the reported accuracy of the tests conducted using the application was low, and 80% of the participants were found to have deviated from the test instructions (i.e., lost count of the number of laps, did not measure the distance walked, or did not walk at their maximum speed). Thus, given the extent of the test deviations reported in this healthy population, the technology-based method assessed in this study appears unlikely to be suitable for use by elderly people or patients with cognitive difficulties, highlighting the need for a more automated and easy-to-use tool.

The results of the test-retest analyses of the FeetMe at home measurements can be compared to those reported for other systems that have been evaluated for self-administering the 6MWT. Brooks et al. [28] evaluated the performance of a smartphone-based application for assessing 6MWTs conducted at home by 19 participants, including patients with congestive heart failure or pulmonary hypertension, and healthy controls. At least three tests with a two-week interval were performed by each participant. Analyses of the results revealed a CV value of 4.7% for the smartphone application, compared to the lower CV value of 3.33% obtained for the FeetMe device used at home in the current study. One of the most promising previous studies evaluating a solution for carrying out 6MWTs remotely was that by Wevers et al. [33]. This study investigated the use of a global positioning system (GPS) by investigators administering 6MWTs to 27 patients with chronic stroke outdoors in the patients' own neighborhoods [33]. A measuring wheel was also used by the investigators as a reference and the official 6MWT guidelines were followed as closely as possible, including the use of a 30-m track. The results obtained for the reproducibility of the GPS-estimated 6MWDs were very good, with an

ICC of 0.96 and an SEM of 18.1 m. Remarkably, the values obtained for the FeetMe device at home in the current study were slightly better for the SEM (15.02 m) and very similar for the ICC (0.95). In addition, although the GPS appeared to provide a well suited and accurate solution for conducting 6MWTs remotely, unlike the FeetMe system, the GPS cannot be used to conduct the tests indoors.

## Study limitations

The current study provided the first assessment of the test-retest reliability of FeetMe insoles for measuring 6MWDs during tests conducted independently by the participants at home and compared it to the reliability of the evaluation in hospital settings. However, this study also had some limitations. In particular, this study was carried out using in-hospital tests conducted at a single center in a single country and involved healthy volunteers rather than patients with pathological gait. Besides, some of the authors of this work were affiliated with FeetMe company. Future studies by an independent group of researchers are therefore required to replicate these results in a larger sample population, including both healthy volunteers and those with gait anomalies, with in-hospital tests conducted at multiple centers and in multiple countries. In addition, the learning effect between repeated 6MWTs at home should be further studied and compared to that in hospital settings [24].

## Conclusions

In conclusion, this study demonstrated that the FeetMe connected insoles provide a reliable solution for allowing 6MWTs to be self-administered independently at home by healthy adults. This finding makes a substantial contribution to the progress of remote patient monitoring, a crucial aspect of future clinical practice. Indeed, at-home monitoring of gait, through 6MWT for instance, would remove the patient burden associated with commuting to hospital assessment centers, and would drastically simplify the patient's care. The home setting would also allow for more frequent assessments of the functional capacity of patients, and therefore result in better patient follow-up and, ultimately, in overall improvements in patient management.

Besides, it is important to mention, that in addition to measuring the 6MWD, the FeetMe device has the capability of collecting additional information on patient gait parameters during the test, providing complementary data, which when analyzed together with the 6MWD, can help obtain a finer understanding of a patient's condition. This makes the FeetMe device one of the most convincing solutions for at-home gait monitoring. Therefore, further studies are endorsed to evaluate its performance across various pathologies, thereby enhancing its potential for widespread clinical application.

## Acknowledgments

The authors would like to thank all volunteers who agreed to participate in this study. We also thank Margarita Arango, Ayelen Gallardo, Christelle Saulnier and prof Caroline Moreau for proofreading of this manuscript and Gilles Monneret for advising on statistical analysis. We are grateful to Cyril Basquin for data management services, and for his input on the study methodology. We would also like to thank Drs Emma Pilling and Marielle Romet (Santé Active Edition–Synergy Pharm) for medical writing assistance and language editing. We express our gratitude to FeetMe company for providing FeetMe insoles for this study.

## Author Contributions

**Conceptualization:** Andrey Mostovov, Damien Jacobs, Leila Farid, Guillaume Baille.

**Formal analysis:** Damien Jacobs.

**Investigation:** Paul Dhellin, Guillaume Baille.

**Methodology:** Andrey Mostovov, Damien Jacobs, Leila Farid, Guillaume Baille.

**Project administration:** Leila Farid, Guillaume Baille.

**Supervision:** Leila Farid.

**Writing – original draft:** Andrey Mostovov, Damien Jacobs.

**Writing – review & editing:** Andrey Mostovov, Damien Jacobs, Guillaume Baille.

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
