## [Decision Letter · Decision Letter 0]

22 Jun 2023

PDIG-D-23-00161

Excellent test-retest reliability of the six-minute walking distance measured by FeetMe® insoles during tests conducted with a one-week interval by completely unassisted healthy adults in their homes.

PLOS Digital Health

Dear Dr. Mostovov,

Thank you for submitting your manuscript to PLOS Digital Health. After careful consideration, we feel that it has merit but does not fully meet PLOS Digital Health's publication criteria as it currently stands. Therefore, we invite you to submit a revised version of the manuscript that addresses the points raised during the review process.

Please submit your revised manuscript within 60 days Aug 21 2023 11:59PM. If you will need more time than this to complete your revisions, please reply to this message or contact the journal office at digitalhealth@plos.org. Please include the following items when submitting your revised manuscript:

We look forward to receiving your revised manuscript.

Kind regards,

Haleh Ayatollahi

Section Editor

PLOS Digital Health

Journal Requirements:

1. Please provide separate figure files in .tif or .eps format only and remove any figures embedded in your manuscript file. Please also ensure that all files are under our size limit of 10MB.

2. We noticed that you used unpublished data in the manuscript. We do not allow these references, as the PLOS data access policy requires that all data be either published with the manuscript or made available in a publicly accessible database. Please amend the supplementary material to include the referenced data or remove the references.

3. We do not publish any copyright or trademark symbols that usually accompany proprietary names, eg ©, ®, ™ (e.g. next to drug or reagent names). Please remove all instances of trademark/copyright symbols throughout the text, including ®.

4. Figure 1 contains screenshots. We are not permitted to publish these under our CC-BY 4.0 license; websites are usually intellectual property and are copyrighted.This includes peripheral graphics of the web browser such as the buttons. We ask that you please remove or replace it.

5. In the online submission form, you indicated that "Data available on request due to restrictions e.g. privacy or ethical". All PLOS journals now require all data underlying the findings described in their manuscript to be freely available to other researchers, either 1. In a public repository, 2. Within the manuscript itself, or 3. Uploaded as supplementary information.

Additional Editor Comments (if provided):

Reviewers' comments:

Reviewer's Responses to Questions

**Comments to the Author**

1. Does this manuscript meet PLOS Digital Health’s publication criteria? Is the manuscript technically sound, and do the data support the conclusions? The manuscript must describe methodologically and ethically rigorous research with conclusions that are appropriately drawn based on the data presented.

Reviewer #1: Yes

Reviewer #2: No

Reviewer #3: Yes

2. Has the statistical analysis been performed appropriately and rigorously?

Reviewer #1: Yes

Reviewer #2: No

Reviewer #3: Yes

3. Have the authors made all data underlying the findings in their manuscript fully available (please refer to the Data Availability Statement at the start of the manuscript PDF file)?

Reviewer #1: No

Reviewer #2: Yes

Reviewer #3: No

4. Is the manuscript presented in an intelligible fashion and written in standard English?

Reviewer #1: Yes

Reviewer #2: Yes

Reviewer #3: Yes

5. Review Comments to the Author

Reviewer #1: Dear authors,

Your submission titled 'Excellent test-retest reliability of the six-minute walking distance measured by

FeetMe® insoles during tests conducted with a one-week interval by completely

unassisted healthy adults in their homes' was an interesting read and was an appropriate submission for PLOS Digital Health. The study evaluated the potential of a wearable smart insole to empower patients with monitoring their gait at their convenience. The results highlight the tested device's potentials in assessing conditions with gait disturbances. 

However, there are some aspects that need to be addressed. While the conflict of interest have been disclosed, a majority of the authors are/have been related to the company providing the device tested, in addition of the financial support provided by the FeetMe company. This requires a measure of caution when drawing conclusions and should be further highlighted within the text, preferably in a dedicated limitations sections.

On a related note, some of the literature used to support the potentials of the solution (14, 15) included authors who were employees of the FeetMe company, hence indicating need for caution regarding conflicting interests. Deriving such potentials from independent studies would add more value to the claims.

Some clarifications on the procedures and results would also be beneficial to add to the body of literature. For example, in the Intervention section, the length of the training of the participants could have been specified. Reasons for why some patients missed taking the required test and the relevant technical issues would also be important to share. 

As the use of a companion smartphone application has been made, its compliance to relevant privacy policies should be specified. The field of digital health is not focused only on the technological component but also integrates the relevant ethical, social and regulatory concerns. As the FeetMe smart insoles is a commercial product, acknowledging its compliance to relevant policies would be a recommendation for scientific publication.

While the authors compared their findings with other research in gait studies, they did not do so with other studies that analysed other smart insoles for similar purposes. There have been a number of such studies in the past 5 years or so which the authors could compare their findings to in order to provide a more objective analysis of their findings.

It could also be recommended for the authors to use more cautious language when deriving recommendations in the discussion and conclusions. This is because the participants were healthy and the generalisability of the findings can be challenging to extrapolate to patients with gait disturbances. Further limiting the generalisability is that only 4 interventions took place for each participant. Thus, it could also be recommended to rephrase the study title as a feasibility assessment or pilot to be more indicative of the study scope.

This concludes my review comments and I hope this feedback is useful for your manuscript going further.

Reviewer #2: Thank you to the authors and editors for the opportunity to review this paper. This paper describes the utility of FeetMe, an innovative technology which may help capture measurements in non-clinical settings. Although I appreciate the goal of this study and the gap in intervention it is aiming to fill, I have strong concerns regarding the methodology and conclusions of the study. I have organized my concerns by section. 

Introduction:

The authors identify a clear gap in measurement and provide a thoughtful solution to help fill this gap. However, clarification is needed throughout this section. 

In line 58, please specify the type of bias and explicitly state how it affects measurement. 

The language used in line 81/82 is misleading, given that the authors are reporting unpublished observations. Please provide more details on this study and explicitly state in text that these results require additional testing and/or peer review. 

In line 84, the authors state that they are assessing reliability and repeatability of the FeetMe. In later sections it seems that they are only assessing test-retest reliability (which can also be called repeatability). Is there another form of reliability being assessed? Additionally, in line 135 the authors state that their primary outcome was to compare the test-retest reliability of at home vs by the rater in the hospital. Please use consistent language and goals throughout the manuscript.

The hypothesis for test-retest reliability, including a targeted kappa score is not provided. This is concerning as it prevents the reader from understanding what the initial goal was and whether or not it was met. 

Methods:

Thank you to the authors for the detailed information in the methods section. 

Please provide details on how participants were recruited. Were all participants recruited through one clinic? This information may help readers understand the context of the study and its generalizability. 

Information on who rated the 6MWT would be useful. 

Thank you for providing information on the soles used. Please provide additional information on if soles were adapted for different feet types/sizes. Additionally, if the soles were ill-fitting, would this affect the results provided? Please describe.

Statistical analysis:

Thank you to the authors for the efforts made in the statistical analysis section of the paper.

The age range included in this study was quite wide. Were there any subgroup analysis made, or any noted differences included across different age ranges? 

I appreciated the use of several statistical tests. For test-retest reliability, kappa statistics to estimate the proportion of observed agreement to the proportion of agreement expected to chance is recommended. Would it be possible to provide the kappa statistic? 

Results:

Thank you to the authors for the detailed results section. 

The authors mention low retest bias in line 179/180, however, the statistical analysis section does not describe how this was calculated, as the units of measurements do not correlate to the above statistical analysis section. Please clarify in the statistical analysis section how this was quantified and calculated. 

Discussion:

Given the small sample size and concerns detected in the methods and statistical analysis section, the final conclusion in the discussion should be reworded. Instead of saying “strong evidence” in line 224, the authors could restate to suggest that FeetMe shows promising results for future exploration, rather than suggesting that the app can be used now. 

In line 175 of the results section, the authors describe a large difference between the FeetMe measurements and clinical measurements. Please provide additional information in the discussion section on why this may have occurred. For example, does this suggest that the app does not correlate well with gold standard clinical measurements, or does it require modifications in assessment? 

When describing the use of new technology in research, it is essential to describe any potential ethical concerns. The authors mention that technical details of the app have already been published, however, it is important to note where the data is stored, for how long it is stored, who has access to the data, and how healthcare professionals access the information. Please include a section discussing the potential ethical concerns of the app.

Reviewer #3: A worthwhile piece of research, I can see the application of this technology within clinical practice and how it might benefit the quality of life for patients. Many thanks for the opportunity to review.

This research could be improved in the following ways.

Structure and flow. A large part of the discussion feels like it is providing the justification for this research and therefore should set the scene for the research within the introduction (lines 227-253)

The figure 2 and figure 3 should be moved below the commentary, which explains the figure which follows next (i.e. move lines 158-160 above figure 2 and move figure 3 below lines 167-171).

I suggest a section title for the statistical analyses, provides a clear distinction from the population demographics, which starts with a short introduction to the statistical analyses section.

Table 1 should be moved lower down the section, where the results within it are described.

Results interpretation, the researchers have highlighted and explained the difference between hospital rater and the FeetMe at home tests, however, have not sufficiently explained or interpreted the results between the FeetMe at home and in hospital results. A further commentary outlining the differences in the results and exploring possible reasons for this should be expanded upon. 

Clearly defined implications for future research (or similar) and learnings from this study sections, would add value to the manuscript as would a section more clearly defined as limitations.

Overall improvements could be made to structure and flow with use of headings to break up longer sections to aid readability.

6. PLOS authors have the option to publish the peer review history of their article (what does this mean?). If published, this will include your full peer review and any attached files.

**Do you want your identity to be public for this peer review?** For information about this choice, including consent withdrawal, please see our Privacy Policy.

Reviewer #1: No

Reviewer #2: No

Reviewer #3: Yes: Shoshana Bloom

---

## [Decision Letter · Decision Letter 1]

12 Sep 2023

PDIG-D-23-00161R1

Excellent test-retest reliability of the six-minute walking distance measured by FeetMe insoles during tests conducted with a one-week interval by completely unassisted healthy adults in their homes.

PLOS Digital Health

Dear Dr. Mostovov,

Thank you for submitting your manuscript to PLOS Digital Health. After careful consideration, we feel that it has merit but does not fully meet PLOS Digital Health's publication criteria as it currently stands. Therefore, we invite you to submit a revised version of the manuscript that addresses the points raised during the review process.

Please submit your revised manuscript within 30 days Oct 12 2023 11:59PM. If you will need more time than this to complete your revisions, please reply to this message or contact the journal office at digitalhealth@plos.org. Please include the following items when submitting your revised manuscript:

We look forward to receiving your revised manuscript.

Kind regards,

Haleh Ayatollahi

Section Editor

PLOS Digital Health

Journal Requirements:

2. We do not publish any copyright or trademark symbols that usually accompany proprietary names, eg ©, ®, ™ (e.g. next to drug or reagent names). Please remove all instances of trademark/copyright symbols throughout the text, including ® on page 17.

Additional Editor Comments (if provided):

The manuscript was interesting; please address the following comments in your next revision.

1- The title is a bit long. Could you please make it shorter?

2- Please follow the journal instructions for the manuscript preparation including the abstract format.

3- Please add appropriate keywords based on the MeSH terms.

4- Please ensure that the aim of the study is the same in the abstract, introduction, etc.

Reviewers' comments:

Reviewer's Responses to Questions

**Comments to the Author**

1. If the authors have adequately addressed your comments raised in a previous round of review and you feel that this manuscript is now acceptable for publication, you may indicate that here to bypass the “Comments to the Author” section, enter your conflict of interest statement in the “Confidential to Editor” section, and submit your "Accept" recommendation.

Reviewer #1: (No Response)

Reviewer #3: All comments have been addressed

2. Does this manuscript meet PLOS Digital Health’s publication criteria? Is the manuscript technically sound, and do the data support the conclusions? The manuscript must describe methodologically and ethically rigorous research with conclusions that are appropriately drawn based on the data presented.

Reviewer #1: Yes

Reviewer #3: Yes

3. Has the statistical analysis been performed appropriately and rigorously?

Reviewer #1: Yes

Reviewer #3: Yes

4. Have the authors made all data underlying the findings in their manuscript fully available (please refer to the Data Availability Statement at the start of the manuscript PDF file)?

Reviewer #1: Yes

Reviewer #3: Yes

5. Is the manuscript presented in an intelligible fashion and written in standard English?

Reviewer #1: Yes

Reviewer #3: Yes

6. Review Comments to the Author

Reviewer #1: Thank you to the authors for addressing the initial feedback. However, there are minor aspects that need to be considered further. 

While the authors have specified the CE/FDA classifications regarding the use of the application, the data handling policies were not specified. Data collection is explicitly mentioned in line 125 where the relevant data handling policy that was adhered to can be specified as good practice for scientific publication.

Regarding the title, considering the limited number of interventions, lack of control groups and the focus on healthy participants (while aiming to address patients with gait issues), it would be advisable to rephrase the study title as a feasibility assessment or pilot study to be more indicative of the study scope.

Reviewer #3: All my comments have been addressed within the revised manuscript. No further comments from me at this time. Many thanks.

7. PLOS authors have the option to publish the peer review history of their article (what does this mean?). If published, this will include your full peer review and any attached files.

**Do you want your identity to be public for this peer review?** For information about this choice, including consent withdrawal, please see our Privacy Policy. 

Reviewer #1: No

Reviewer #3: Yes: Shoshana Bloom

---

## [Decision Letter · Decision Letter 2]

16 Oct 2023

Test-retest reliability of the six-minute walking distance measurements using FeetMe insoles by completely unassisted healthy adults in their homes.

PDIG-D-23-00161R2

Dear Dr Mostovov,

We are pleased to inform you that your manuscript 'Test-retest reliability of the six-minute walking distance measurements using FeetMe insoles by completely unassisted healthy adults in their homes.' has been provisionally accepted for publication in PLOS Digital Health.

Best regards,

Haleh Ayatollahi

Section Editor

PLOS Digital Health

Reviewer Comments (if any, and for reference):

Reviewer's Responses to Questions

**Comments to the Author**

1. If the authors have adequately addressed your comments raised in a previous round of review and you feel that this manuscript is now acceptable for publication, you may indicate that here to bypass the “Comments to the Author” section, enter your conflict of interest statement in the “Confidential to Editor” section, and submit your "Accept" recommendation.

Reviewer #1: All comments have been addressed

2. Does this manuscript meet PLOS Digital Health’s publication criteria? Is the manuscript technically sound, and do the data support the conclusions? The manuscript must describe methodologically and ethically rigorous research with conclusions that are appropriately drawn based on the data presented.

Reviewer #1: Yes

3. Has the statistical analysis been performed appropriately and rigorously?

Reviewer #1: Yes

4. Have the authors made all data underlying the findings in their manuscript fully available (please refer to the Data Availability Statement at the start of the manuscript PDF file)?

Reviewer #1: Yes

5. Is the manuscript presented in an intelligible fashion and written in standard English?

Reviewer #1: Yes

6. Review Comments to the Author

Reviewer #1: Dear authors,

Thank you for addressing the rounds of feedback and providing clarifications where needed. I am happy to recommend the publication of your research.

7. PLOS authors have the option to publish the peer review history of their article (what does this mean?). If published, this will include your full peer review and any attached files.

**Do you want your identity to be public for this peer review?** For information about this choice, including consent withdrawal, please see our Privacy Policy.

Reviewer #1: No
